# Efficiency of Nisin as Preservative in Cosmetics and Topical Products

**Elisabete Maurício** [1,2,3,*] (iD)**, Catarina Rosado** [1] (iD)**, Maria Paula Duarte** [4]**, Joana Verissimo** [2]**, Sara Bom** [3] **and Laura Vasconcelos** [3]

[1] CBIOS-Universidade Lusófona Campo Grande, 376, 1749-024 Lisboa, Portugal; catarina.rosado@ulusofona.pt

[2] Elisa Câmara, Lda, Dermocosmética, Talaide, 2785-601 São Domingos de Rana, Portugal; geral@elisacamara.pt

[3] Faculdade de Engenharia, Universidade Lusófona, Campo Grande, 376, 1749-024 Lisboa, Portugal; sarabom@campus.ul.pt (S.B.); vlaura5@gmail.com (L.V.)

[4] Metrics/DCTB, Faculdade de Ciências e Tecnologia, Universidade Nova de Lisboa, 2829-516 Caparica, Portugal; mpcd@fct.unl.pt

* Correspondence: elisabetem@elisacamara.pt; Tel.: +351-21-751-5500 (ext. 627)

**Abstract:** Nisin is a bacteriocin synthesized by certain species of *Lactococcus lactis*, that has been recently employed as a preservative in the food industry. Taking into account its potential as a natural preservative, its applicability in cosmetics and topical products was probed, aiming to replace or reduce the use of synthetic preservatives currently used in these products. In vitro susceptibility tests were performed using the plate diffusion method and the "Challenge Test". The action of nisin was tested when applied alone and in synergy with ethylenediaminetetraacetic acid tetrasodium salt (EDTA) and similar synthetic preservatives, Abiol® (INCI-Imidazolidinyl urea) and Microcare PM2 (Phenoxyethanol, Ethylparaben, Methylparaben). The results of this study demonstrate that nisin is effective in inhibiting gram-positive microorganisms *Staphylococcus aureus* and *Bacillus* sp. However, for other tested microorganisms, only the combination of nisin, EDTA and synthetic preservatives, respectively at 125 ppm/0.1/0.35%, showed antimicrobial activity in compliance with criterion A from ISO 11930. With this study, it is concluded that nisin can be a viable alternative when associated with other preservatives, reducing the use of higher doses of chemical/synthetic preservatives that are often associated with sensitivity and allergic reactions.

**Keywords:** nisin; preservative; cosmetics; bactericide; antimicrobial tests

## 1. Introduction

In recent years, one of the challenges of the cosmetic industry is the application of natural preservatives in its products, in order to avoid possible allergic reactions and sensitivity phenomena that are often associated with the use of chemical preservatives [1,2].

A preservative is a synthetic/chemical or natural ingredient with antimicrobial properties that is added to cosmetics and personal care products to maintain microbiological quality. Preservatives are used to increase the shelf life of these formulations, inhibiting the growth of microorganisms. Thus, they prevent the deterioration caused by bacteria (Gram-positive and Gram-negative), moulds and yeasts that can cause disease or simply disrupt the smooth appearance of the final product. A good preservative should provide broad-spectrum activity, i.e., eliminate all types of microorganisms [1–4].

The fundamental property of any preservative is that it should be effective at low concentrations and not toxic for humans [5]. The "ideal" preservative for use in cosmetics should be soluble in water and stable at any temperature and pH conditions that are used in the manufacturing process.

The preservatives should be also colorless and odorless, or should not add color or odor to the product, and should not react with other ingredients [1–4,6]. Combinations of preservatives (blends) could be a good solution to reduce the possible side effects associated with the chemical preservation method [5,7–9].

Nisin, a natural preservative structurally composed of 34 amino acid residues (3500 Da), is biosynthesized by certain species of *Lactococcus lactis* subsp. *lactis* during the exponential phase of growth of the microorganism. It belongs to the family of lantibiotics (linear lantibiotic), a group of antimicrobial peptides (bacteriocins) produced by a wide range of Gram-positive bacteria [6,10–12]. Bacteriocins are proteins or complexes of proteins that act against Gram-positive bacteria (acting in the peptidoglycan walls), including lactic acid and sporulated bacteria [13], but with low activity against Gram-negative bacteria, moulds and yeasts. Gram-negative bacteria are resistant because of their outer membranes, however, the addition of a chelating agent such as EDTA, can destabilize these membranes, making cells more sensitive to nisin [6,14].

Nisin has different natural variants. Nisin A was the first isolated form of nisin, but up to now a further five natural variants (nisin Z, F, Q, U and U2) have been described, which differ by up to 10 amino acids. *Lactococcus lactis*, produces the variants A, Z, F and Q while *Streptococcus* sp. produces the variants U and U2 [15]. Additionally, bioengineering strategies have been applied to develop nisin derivatives with enhanced activity against specific pathogens, including gram-negative bacteria [16].

The oral lethal dose ($LD_{50}$) in rats of purified nisin was more than 2000 mg/kg body weight and the $LD_{50}$ for nisin preparation in mice was 6950 mg/kg body weight. Moreover, nisin is not carcinogenic or mutagenic and is not associated with any reproductive or developmental toxicity [17]. This bacteriocin was considered GRAS "Generally Regarded as Safe" and accepted by the *Codex Alimentarius* as a food preservative (Food and Agriculture Organization) [18], being applied mainly in the dairy industry [18,19]. EFSA issued an opinion on the use of nisin as a food additive [20]. The Panel did not find any data, that would warrant any change of the Acceptable Daily Intake (ADI) of 0.13 mg nisin/kg bw previously established by Scientific Committee on Food (SCF). The Panel furthermore noted that there has been exposure to nisin for many centuries and concluded that nisin can be safely used.

For industrial purposes, its solubility and stability increase with the acidity of the medium, being stable at pH 2, and can be autoclaved at 121 °C without losing any of its properties [19]. Besides its application in the food industry, nisin can also be applied in many other fields, such as the medical-pharmaceutical, chemical, cosmetic, among others [6,21–23].

Taking into account the potential use of this product as a natural preservative, a study was carried out to evaluate the antimicrobial potential of nisin towards some of the microorganisms more susceptible to contaminate cosmetic products: *Pseudomonas aeruginosa*, *Staphylococcus aureus*, *Candida albicans*, *Escherichia coli*, *Aspergillus brasiliensis* as suggested in ISO 11930 (2012) [24]. Additionally, *Bacillus* sp. was also tested, because of its spore's resistance in an industrial environment. The main and fundamental aim of this study is to replace or reduce the current storage system based on synthetic preservatives.

## 2. Materials and Methods

This study evaluated the efficacy of nisin from *L. lactis*, Sigma-Aldrich, St. Louis, MO, USA (a commercial preparation of 2.5%—balance sodium chloride and denatured milk solids, CAS 1414-45-5) at several concentrations and when added with the chemical preservatives Abiol®-3V Sigma s.p.a., Bergamo, Italy, (INCI-Imidazolidinyl urea); Microcare PM2®, Thor, France (INCI-Phenoxyethanol, Ethylparaben, Methylparaben); and Tetrasodium EDTA, chelating agent, Esperis s.p.a., Milan, Italy.

### 2.1. Microorganisms Used

The microorganisms used in these tests were from the ATCC collection of the Elisa Câmara laboratory. The *Bacillus* sp. was isolated from industrial contaminated cosmetics, Elisa Câmara, Lda.

To prepare the working culture of the microorganisms, a subculture was made from the stock culture in plates with recommended media for each microorganism, according to strains suppliers and ISO 11930 (2012) [24]. Table 1 shows the microorganisms, incubation conditions and culture media under study.

**Table 1.** Microorganisms used in the study; culture media and incubation conditions.

| Microorganisms | Culture Media Oxoid, U.K. | Incubation Conditions (Time/Temperature) |
|---|---|---|
| *Aspergillus brasiliensis* ATCC16404 | Potato dextrose agar | 5–7 days/25 °C |
| *Bacillus* sp. EC A02 | Tryptic soy agar | 24 h/37 °C |
| *Candida albicans* ATCC10231 | Sabouraud dextrose medium agar | 24 h/37 °C |
| *Escherichia coli* ATCC8739 | Tryptic soy agar | 24 h/37 °C |
| *Pseudomonas aeruginosa* ATCC9027 | Tryptic soy agar | 24 h/37 °C |
| *Staphylococcus aureus* ATCC25992 | Tryptic soy agar | 24 h/37 °C |

## 2.2. Tested Product

The cosmetic base formulation tested was a royal jelly milk (O/W emulsion) provided by Elisa Câmara® Lda, pH 5.5. Ingredients (INCI): Aqua, Caprylic/Capric Trygliceride (Waglinol® 3/9280; IQL-Industrial Química; Lasem S.A.U, Barcelona, Spain); Cetearyl Alcohol, Sodium Cetearyl Sulfate (Galenol® 16 18 CS; Sasol GmbH, Hamburg, Germany); Polyglyceryl-3 Methylglucose Distearate (Tego Care® 450; Evonik nutrition and care GmbH, Essen, Germany) Propylene Glycol (Monopropilenoglicol USP; RNM-produtos químicos, Famalicão, Portugal); Glycerin (Glycerine USP, PH. Eur.; Interfat-natural oils, Barcelona, Spain); Panax Ginseng extract (Extrapone® Ginseng special; Symrise GmbH & Co. KG, Holzminden, Germany); Royal Jelly (Lipoplastidine® Pappa regalis, Vevy Europe s.p.a., Genova, Italy); Parfum (FLORAL® No. 54.690.0225, Sensient Fragrances, S.A., Granada, Spain; BHT (Purolan® BHT; LANXESS Distribution GmbH, LeverKusen, Germany).

## 2.3. In Vitro Sensitivity Test by Disk Diffusion Method

The conventional disk diffusion method was employed for the initial assessment of antimicrobial potential [25]. A saline bacterial suspension, with 1% triptone USP (Biokar, Allone, France) and 0.85% sodium chloride (Panreac, Barcelona, Spain), corresponding to $1$–$2 \times 10^8$ CFU/mL for bacteria, $1$–$5 \times 10^6$ CFU/mL for yeasts and $0.4$–$5.5 \times 10^6$ CFU/mL for moulds [26,27], was prepared and used to inoculate the Muller Hinton Agar plates (Biokar, Allone, France). Six paper discs (6 mm in diameter) were impregnated with 50 µL of antimicrobial solutions under test. After 24 h incubation the formation of inhibition zones of growth was observed. The antimicrobial activities were determined by measuring the diameter of the inhibition zone.

Solutions of nisin and other synthetic preservatives were prepared in water or propylene glycol, according to their solubility. A positive and a negative control were prepared. The following formulations were tested: nisin 125 ppm; 250 ppm; 500 ppm and 1000 ppm; nisin 125 ppm; 250 ppm; 500 ppm and 1000 ppm with 0.1% EDTA; nisin 125 ppm; 250 ppm; 500 ppm and 1000 ppm with 0.15% Abiol and 0.35% Microcare PM2; 0.15% Abiol and 0.35% Microcare PM2; 0.30% Abiol and 0.70% Microcare PM2 (positive control); sterile water (nisin negative control); Propylene glycol (synthetic preservatives negative control).

## 2.4. Challenge Test (ISO 11930 (2012))

The efficacy of preservatives was tested by the challenge test following the ISO 11930 standards (2012) [24], which determines the amount of preservative that ensures product effective preservation by studying the reduction of the microorganisms over 28 days. All culture media and diluents were

prepared following the manufacturer's instructions and the microorganisms used in the study were cultured in ideal conditions (Table 1).

Several cosmetic milk emulsions were prepared using the previously described base formulation (see Section 2.2), with different concentrations/combinations of preservatives: nisin 1000 ppm; nisin 250 ppm with 0.1% of EDTA; nisin 125 ppm with 0.15% Abiol and 0.35% Microcare PM2; 0.15% Abiol and 0.35% Microcare PM2; formulation without preservative (negative control); 0.3% Abiol and 0.7% Microcare PM2 (positive control).

Then, the different formulations prepared (samples of 20 g) were introduced in sterile containers, separately inoculated with each one of the six strains and then observed for 28 days at 25 °C.

Colony counts were performed at contact times of 0, 1, 7, 14, 21 and 28 days, using the plate count method of ISO 11930 (2012) [24]. The results were expressed as log CFU $g^{-1}$. All determinations were performed in triplicate.

The preservation of the formulation was considered effective if the formulation meets criteria A and B, respectively:

**Criterion A**—the total number of bacteria is reduced in two to three logarithmic units between the second and the seventh day in the challenge test, and an increase in population doesn't occur to the end of the test (28th day). For fungi, the criterion is met when the total number is reduced in two logarithmic units until the 14th day, without any growth up to the last day. The cosmetic product is protected against microorganisms that may be a potential hazard for the consumer's health if this criterion is fulfilled by the formulation for all recommended microorganisms.

**Criterion B**—the total number of bacteria is reduced in three logarithmic units until the 14th day without any growth up to the 28th day. For fungi, the criterion is met when the total number is reduced in one logarithmic unit until the 14th day, without any growth up to the last day. The formulation fulfills the standards if it meets this criterion, however, additional risk analysis is recommended, demonstrating the existence of control factors not related to the formulation, like special packaging material.

In the cases where the formulation doesn't respect criteria A or B, it is exposed to high risk of contamination. A complete risk analysis study has to be performed in order to avoid potential contaminations during the product's lifetime, or otherwise the product should be reformulated with new preservatives to ensure its security.

## 3. Results

The methodology used in this study was based on sensitivity assays using the disk diffusion method and the Challenge Test and intends to determine the efficiency of nisin as a cosmetic preservative either individual or in association with synthetic preservatives.

A preliminary efficacy assessment was made using the disk diffusion method. In this assay the disks were impregnated with different concentrations of nisin or with nisin and several preservatives combinations, in order to assess the best concentrations and its efficacy as preservative in a cosmetic or topic formulation. The antimicrobial effect was detected by the formation of an inhibition zone around the disk. Results obtained in this assay were thereafter confirmed with the Challenge Test, which assesses the efficacy of the preservative system with greater precision and accuracy.

Through the analysis of Table 2 it is possible to observe that when applied alone, nisin presented antimicrobial efficacy against Gram-positive bacteria (*S. aureus* and *Bacillus* sp.). On the other hand, it was found to be ineffective for the remaining microorganisms studied, namely Gram-negative bacteria, moulds and yeasts. These results are in agreement with other authors, who observed the antimicrobial power of this compound against Gram-positive bacteria [1–4,6,8], and a less significant effect, or even no effect on the growth of Gram-negative bacteria, moulds and yeasts [6,10,12,14,21].

**Table 2.** Results of disk diffusion assay for nisin and other preservatives. (+) inhibition zones—between 10 and 20 mm; (-) inhibition zones <10 mm—inefficiency of antimicrobials tested.

| Tested Preservatives | *Aspergillus brasiliensis* | *Bacillus* sp. | *Candida albicans* | *Escherichia coli* | *Pseudomonas aeruginosa* | *Staphylococcus aureus* |
|---|---|---|---|---|---|---|
| Nisin 125 ppm | - | + | - | - | - | + |
| Nisin 250 ppm | - | + | - | - | - | + |
| Nisin 500 ppm | - | + | - | - | - | + |
| Nisin 1000 ppm | - | + | - | - | - | + |
| Nisin 125 ppm + EDTA 0.1% | - | + | - | - | - | + |
| Nisin 250 ppm + EDTA 0.1% | - | + | - | - | - | + |
| Nisin 500 ppm + EDTA 0.1% | - | + | + | + | - | + |
| Nisin 1000 ppm + EDTA 0.1% | - | + | + | + | - | + |
| Nisin 125 ppm + Abiol 0.15% + Microcare PM2 0.35% | + | + | + | + | + | + |
| Nisin 250 ppm + Abiol 0.15% + Microcare PM2 0.35% | + | + | + | + | + | + |
| Nisin 500 ppm + Abiol 0.15% + Microcare PM2 0.35% | + | + | + | + | + | + |
| Nisin 1000 ppm + Abiol 0.15% + Microcare PM2 0.35% | + | + | + | + | + | + |
| Abiol 0.15% + Microcare PM2 0.35% | - | - | + | + | - | - |
| Abiol 0.3% + Microcare PM2 0.7% | + | + | + | + | + | + |
| Sterile water | - | - | - | - | - | - |
| Propylene glycol | - | - | - | - | - | - |

Association of nisin with EDTA increased its antimicrobial efficacy (for *C. albicans* and *E. coli*), an improvement that can be seen in the inhibitory effect of 500 and 1000 ppm of nisin (Table 2), but, once again no inhibition of all the microorganisms under study could be observed.

In turn, the association of nisin with EDTA and synthetic preservatives Abiol and Microcare PM2 showed to be effective in inhibiting all microorganisms, even when concentrations of synthetic preservatives applied (0.15% Abiol + 0.35% Microcare PM2) were lower than those recommended by the suppliers for proper conservation of cosmetic products (0.3% Abiol + 0.7% PM2).

The efficacy of nisin as a cosmetic preservative was further assessed by the Challenge Test. In this test nisin was assayed alone or in combinations with just EDTA or EDTA and synthetic preservatives (Figures 1–3).

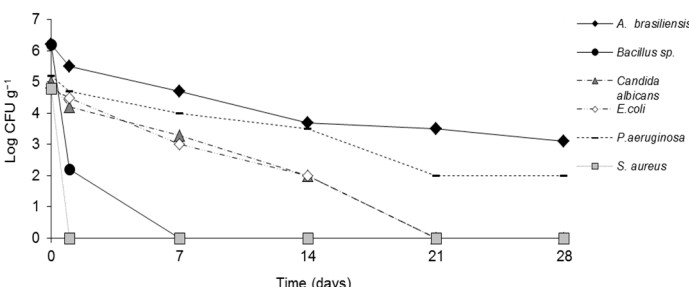

**Figure 1.** Growth inhibition of different microorganisms in body milk formulation with preservative system nisin 1000 ppm.

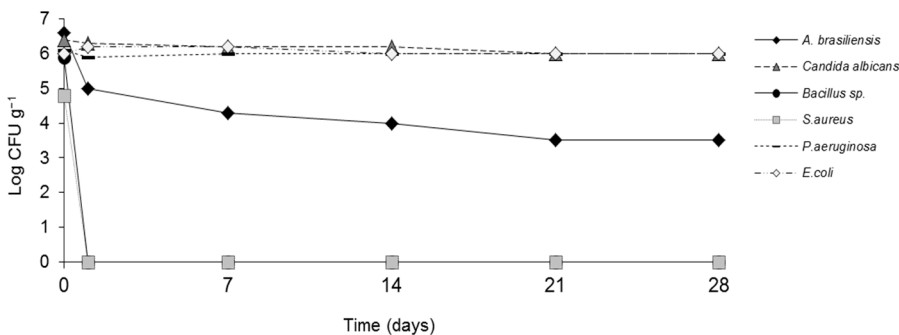

**Figure 2.** Growth inhibition of different microorganisms in body milk formulation with preservative system nisin 250 ppm + 0.1% EDTA.

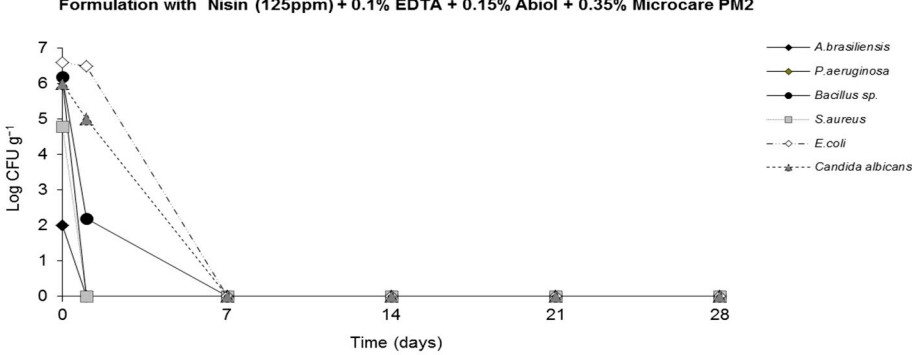

**Figure 3.** Growth inhibition of different microorganisms in body milk formulation with preservative system nisin 125 ppm + 0.1% EDTA + 0.15% Abiol + 0.35% Microcare PM2.

The positive control (0.3% Abiol + 0.7% Microcare PM2), the association with lower concentrations of synthetic chemical preservatives (0.15% Abiol + 0.35% Microcare PM2) and the negative control (formulation without any preservative) were also tested, in order to validate the results (Figures 4–6).

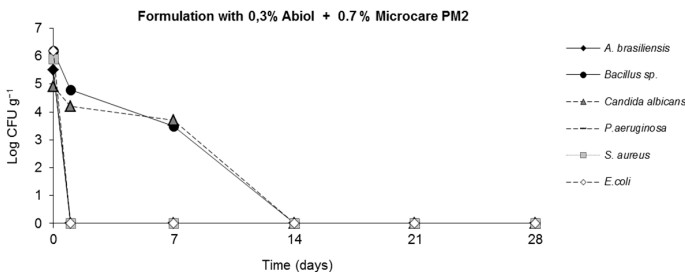

**Figure 4.** Growth inhibition of different microorganisms in body milk formulations with synthetic preservative system 0.3% Abiol + 0.7% Microcare PM2 (positive control).

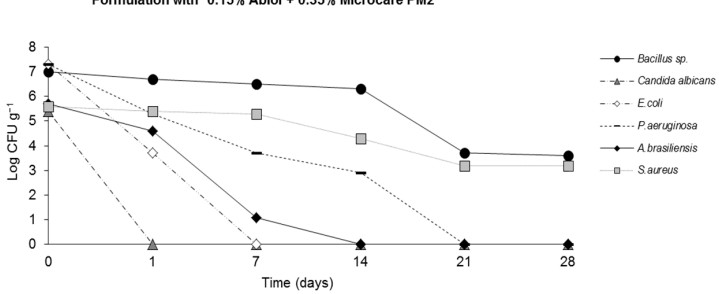

**Figure 5.** Growth inhibition of different microorganisms in body milk formulations with synthetic preservative system 0.15% Abiol + 0.35% Microcare PM2.

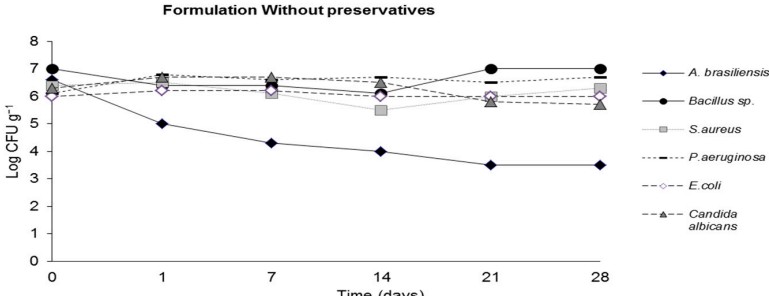

**Figure 6.** Growth inhibition of different microorganisms in body milk formulations without preservatives (negative control).

Results obtained in the Challenge Test showed that the addition of nisin, both individually and in association with synthetic chemical preservatives, in a topical cosmetic formulation was very efficient against the Gram-positive bacteria assayed (*S. aureus* and *Bacillus* sp.). As observed in the disk diffusion assay, in the Challenge Test both Gram-positive bacteria were more sensitive to nisin than to the lower dose of synthetic preservatives tested (0.15% Abiol + 0.35% Microcare PM2) (Figures 1–5).

It is also possible to observe in Figure 1 that the addition of nisin at 1000 ppm was not efficient in reducing all the microorganisms that were establish in the sample, not meeting the reduction criteria A and B required by the European Standard. Similar results were observed in the combination of EDTA with nisin (Figure 2), which was also unable to reduce the Gram-negative bacteria and fungi.

However, in Figure 3 it is possible to verify that the blend of nisin with EDTA and lower concentrations of synthetic preservatives was able to significantly reduce the microbial population, in compliance with the efficacy criterion A of European Standard. In fact, this combined system led to the total elimination of all microorganisms after seven days of testing, an even faster eradication that than observed with the higher dose of synthetic preservatives (Figure 4).

## 4. Discussion

Skin care products are major contributors to cosmetic allergic contact dermatitis, followed by hair care and nail care products [28]. Preservatives are indispensable in cosmetic products to prevent the growth of microorganisms and subsequent spoilage. However, despite being usually nontoxic, together with fragrances, they are among the most common causes of contact dermatitis [29]. Among the top 10 allergens listed by the 2011/2012 patch test results by the North American Contact Dermatitis Group (NACDG), 3 preservatives ranked No. 4 (Methylchloroisothiazolinone/ methylisothiazolinone-MCI-MI), No. 5 (quaternium-15) and No. 7 (formaldehyde), according to significance-prevalence index number [30]. Recent regulations by the European Commission have banned MCI-MI in all leave-on body products as of July 2015 [31].

Parabens are the most well-known preservatives and have been under the attention of consumers for some time. Even though their safety profile has been more thoroughly studied than for many other types of preservatives, innumerous brands of cosmetics have based part of their marketing strategies in the claim "Paraben-free". Nevertheless, since April 2015, the European Commission has limited the maximum concentration of propylparaben and butylparaben, from the previously allowed limit of 0.4% when used individually and 0.8% when mixed with other esters, to 0.14%, when used individually or together [30]. They were also banned from leave-on products designed for the nappy area of young children below the age of three.

It is, thus, recognizable that there is a very high demand for new, safer preservatives. Another possible strategy is to find new agents that decrease the doses of the currently used synthetic preservatives.

The growing dissemination of scientific studies have encouraged the substitution of synthetic preservatives by natural ones, claiming skin health and total absence of side effects [32,33]. Essential oils and plant extracts are the most used antimicrobial agents. Some examples are *Lavandula officinalis*, *Rosmarinus officinalis*, *Thymus vulgaris*, *Eucalyptus globulus*, *Laurus nobilis*, *Salvia officinalis*, *Melaleuca alternifolia*, *Lonicera caprifoleum* and *Lonicera japonica*. Each has shown to exhibit excellent antimicrobial properties and the possibility to be incorporated into topical formulations as active ingredients or as effective preservatives [34–39]. Nevertheless, high concentrations of essential oils are generally required to confer microbial purity to the products and their application may be limited by changes in organoleptic and textural quality of the products, as well as by causing irritation and allergies in users [40]. These issues are behind the reasons why until the EU has not accepted to date these ingredients as preservatives.

Nisin has been reported to be a natural antimicrobial that can be used in preserving food, essentially by protecting against Gram-positive bacteria. However, other applications of nisin have also been studied. Tong et al. [41] showed that the association of nisin with sodium fluoride exerted a high bactericidal effect on *Streptococcus mutants*, a primary cariogenic pathogen, suggesting that the incorporation of nisin in toothpaste and mouthwash can help in the prevention of dental caries. The attachment of nisin to hyaluronic acid allowed the production of a biopolymer with high antibacterial activity under solution or gel form, which could play an important role in the prevention of bacterial contamination in several applications as wound dressings, contact lenses or cleaning contact lenses solutions [42]. On the other hand, the formulation of topical products with nisin demonstrated potential to be use as an alternative to common antibiotics in the treatment of *S. aureus* infections in atopic dermatitis [43]. However, to the best of our knowledge, nisin has never been studied as a cosmetic preservative.

In this study the challenge assay was conducted according to the ISO 11930 (2012) [24], developed by ISO technical committee ISSO/TC 217, cosmetics, to evaluate the antimicrobial protection of a cosmetic product. According to this standard, the product is separated out into several containers, each being challenged with one microorganism, and is fundamental to determine the sensitivity of each of the microorganisms to the antimicrobial substances used in the formulations. However, other methods recommend the use of either pure or mixed culture suspensions. The utilization of mixed species cultures could be an interesting complement to the results obtained in this study, since it enables

evaluating if this association promotes a different response to an antimicrobial treatment, as reported with mixed species biofilms [44].

In this study the combination of nisin, EDTA and synthetic preservatives (Abiol and Microcare PM2), respectively at 125 ppm/0.1/0.35%, was able to inhibit the growth of different microorganisms in a body milk formulation, showing antimicrobial activity in compliance with criterion A from ISO 11930 (2012) [24]. Therefore, the association of nisin with these preservatives enabled a decrease in the concentration of synthetic preservatives in the final cosmetic formula.

It has been observed that mixtures of different synthetic preservatives with natural antimicrobial substances can improve the synergy between them and have a larger spectrum of activity against the growth of microorganisms in cosmetics. For example, synergism between essentials oils or extracts with parabens and imidazolidinyl urea has been reported [35,45]. Moreover, the association of natural products with chelating substances, such as EDTA, also proved to be very efficient. Besides enhancing the effect of some synthetic preservatives, such as the parabens, imidazolidinyl urea and tert-butyl hydroxyanisole, the addition of EDTA can also protect and stabilize the final formulations [46].

However, the introduction of natural preservatives into topical formulations has to be appropriately studied and validated, to avoid reactions and degradation on the final product, with the consequent reduction of its antimicrobial action leaving the formulation unprotected and damaged. Thus, long-term stability studies of formulations containing nisin should be conducted. Further studies should also be performed to evaluate with precision the best proportions of preservative association, in order to examine the lower concentration with antimicrobial efficacy that simultaneously shows the best efficiency, costs and safety. Moreover, the safety of such formulations is an aspect that also needs to be assessed.

Additionally, other natural substances with antimicrobial activity should be investigated and tested in association with nisin, in order to be possible to replace entirely the use of synthetic preservatives in formulations to promote healthier cosmetic products. Finally, it is important to assure that each formula is well preserved before going into the market, to guarantee microbiological purity under its use and storage.

## 5. Conclusions

In conclusion, results obtained showed that nisin could be used as a natural antimicrobial agent in cosmetic and topical products, particularly when associated with other preservatives. It is an interesting alternative because it enables a decrease in the doses of the chemical preservatives that are often associated with sensitivity and allergic reactions in consumers.

**Author Contributions:** Elisabete Maurício and Laura Vasconcelos conceived and designed the experiments; Elisabete Maurício, Joana Verissimo and Sara Bom performed the experiments; Elisabete Maurício, Maria Paula Duarte and Catarina Rosado analyzed the data; Elisabete Maurício, Maria Paula Duarte and Catarina Rosado wrote the paper.

**Conflicts of Interest:** The authors declare no conflict of interest.

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
