# Peer review of "Efficiency of Nisin as Preservative in Cosmetics and Topical Products"

_cosmetics, doi:10.3390/cosmetics4040041_

Round 1
Reviewer 1 Report
Dear Authors,
the work is well done and it represents a good starting point for further study concerning preservatives in cosmetics.
I would like to know if you have already considered the long term stability of formulations containing nisin. If yes please provide to add some results in the paper.
Author Response
The long term stability of formulations containing nisin has not yet been evaluated. But, in fact, stability is an aspect that needs to be assessed. In this way a phrase was introduced in the discussion to highlight this aspect:
“However, the introduction of natural preservatives into topical formulations has to be appropriately studied and validated, to avoid reactions and degradation on the final product, with the consequent reduction of its antimicrobial action leaving the formulation unprotected and damaged. In this way, long term stability of formulations containing nisin should be evaluated.”
Reviewer 2 Report
Preservatives (decrease and/or replacement) in cosmetics formulation is one of the most important challenge in the cosmetic field. Here, the authors used nisin a well-known natural antimicrobial peptides. The works is interesting due to the challenge test but the discussion on the results is poor and what about side effect of this new formulation?
Questions:
- Q1: Why did you used nisaplin (not mention under this name in the article) and not pure nisin to really study the impact and effect of nisin molecule?
- Q2: In table 2 there is no result on the association of nisin with EDTA and synthetic preservatives but page 5 you mention these results?
- Q3: Could you explain why in your work there is no activity of nisin with EDTA agains P.aeruginosa? because as you write page 2, it's well known that addition of EDTA to nisin solution lead to an antibacterial activity of nisin in solution (as demonstrated for example in the work of Lequeux et al. (ref 36 of your article)). Furthermore could you explain why association of nisin and synthetic molecule have an activity against P. aeruginosa and A. Brasiliensis (table 2) even if they have no activity separetely?
- Q4: In your challenge test, you observe a decrease of bacterial population with a nisin concentration of 1000 ppm. Could you explain this decrease for bacteria against which nisin has not demonstrated any activity?
Author Response
In fact, the side effect of this new formulation is an aspect that needs to be assessed. In this way a phrase was introduced in the discussion to highlight this aspect:
“Therefore, further studies should be performed to evaluate with precision the best proportions of preservative association in order to examine the lower concentration with antimicrobial efficacy that simultaneously shows the best efficiency and costs and safety. Moreover, the safety of this new formulation is an aspect that also needs to be assessed.”
Questions:
- Q1: Why did you used nisaplin (not mention under this name in the article) and not pure nisin to really study the impact and effect of nisin molecule?
Q1: We used a commercial preparation of nisin from Lactococcus lactis from Sigma CAS Number:1414-45-5. This preparation is easy available, well characterized and used in several scientific papers to evaluate nisin properties.
(Mataraci, E., & Dosler, S. (2012). In vitro activities of antibiotics and antimicrobial cationic peptides alone and in combination against methicillin-resistant Staphylococcus aureus biofilms. Antimicrobial agents and chemotherapy, 56(12), 6366-6371.
Mohamed, M. F., Hamed, M. I., Panitch, A., & Seleem, M. N. (2014). Targeting methicillin-resistant Staphylococcus aureus with short salt-resistant synthetic peptides. Antimicrobial agents and chemotherapy, 58(7), 4113-4122.
The purpose of this work was to explore the applicability of a preservative that is avaliable in the market and that is used in the food industry. In future studies, it will be interesting to compare its performance with that obtained by pure nisin.
- Q2: In table 2 there is no result on the association of nisin with EDTA and synthetic preservatives but page 5 you mention these results?
Q2: In table 2 there is no result on the association of nisin with EDTA and synthetic preservatives because this specific association was nor tested by the disk diffusion assay. However this association was tested in challenge test and the results were presented in Figure 3. The results mentioned on page 5 refer to challenge test results. To clarify this point the phrase in page 5 was changed to:
“However, in Figure 3 it is possible to verify that the blend of nisin with EDTA and lower concentrations of synthetic preservatives was able to significantly reduce the microbial population, in compliance with the efficacy criteria A of European Standard.”
- Q3: Could you explain why in your work there is no activity of nisin with EDTA agains P.aeruginosa? because as you write page 2, it's well known that addition of EDTA to nisin solution lead to an antibacterial activity of nisin in solution (as demonstrated for example in the work of Lequeux et al. (ref 36 of your article)). Furthermore could you explain why association of nisin and synthetic molecule have an activity against P. aeruginosa and A. Brasiliensis (table 2) even if they have no activity separetely?
The difference verified between our results and results from Lequeux et al. (2014) could be related with the different experimental procedures applied. In our study the effect was evaluated by disk diffusion assay and in the work of Lequeux et al. (2014) by enumeration of culturable bacteria after 24 h of incubation at 37 ºC in 3 mL of saline buffer. Moreover differences in P. aeruginosa strains and concentrations of EDTA may also contribute to the observed differences.
Although when separately tested neither nisin nor synthetic molecules were effective against P. aeruginosa and A. Brasiliensis when all these compounds were tested in association it was possible to observe an antimicrobial effect against these microorganisms. This result may be due to a synergistic effect among the various compounds. As stated in the discussion section, it has been observed that mixtures of different synthetic preservatives with natural antimicrobial substances may give rise to synergies between them and have a larger spectrum of activity against the growth of microorganisms.
- Q4: In your challenge test, you observe a decrease of bacterial population with a nisin concentration of 1000 ppm. Could you explain this decrease for bacteria against which nisin has not demonstrated any activity?
Q4: In fact there was a decrease in some microorganisms for which nisin shows no antimicrobial effect. However, this reduction has not reached the levels required by criteria A and B of the Standard, and nisin cannot therefore be considered to have a preservative effect against these organisms
Reviewer 3 Report
Overall the manuscript is interesting.
Please see in the PDF all my comments and corrections.
The major issues that need to be taken into consideration are:
1. There are some regulatory issues that are not discussed (e.g EFSA opinion about nisin, essential oils as preservatives in EU) and is recommended to be included.
2. Nisin looks to have several derivatives. Which one has been used?
3. Inhibition zones should be given in supplementary data.
4. There is an issue of the concetration about yeasts’ and moulds’ inoculums.
5. All references should be of the same style.
See all relative comments on the PDF.

Author Response
The manuscript was reviewed according to the comments and corrections presented in the PDF, namely in what concerns to the references style and the concentration of yeast and moulds inoculums. The EFSA opinion about the use of nisin as food additive and the EU stance on the use of essential oils as preservatives were included in the introduction and discussion sections, respectively.
About the variant of nisin used (A (A, Z, Q, F, U), we tested a commercial preparation of nisin from Lactococcus lactis from Sigma CAS Number:1414-45-5 and the supplier does not provide this information. However, as it is produced by Lactococcus lactis, it is not nisin U nor U2, and it is not a nisin derivative.
About Challenge test:
The ISO 11930 is a standard to evaluate the antimicrobial protection of a cosmetic product. This standard was developed by ISO technical committee ISSO/TC 217, cosmetics. Many other methods and protocols have been established and can be performed, for example EU and US pharmacopeia, the test from the CTFA Microbiology Guidelines as well as many others. Furthermore to making safe products the companies and industry need a standard guidance to ensure that tests on cosmetics are carried out in any part of the world and that results from different laboratories are comparable. In this way, we chose to carry out our study according to ISO11930. But this is only a preliminary study, and in the future other methods can be performed, to compare results and to further discuss them.